# Multi-Scale Vehicle Detection for Foreground-Background Class Imbalance with Improved YOLOv2

**DOI:** 10.3390/s19153336

**Published:** 2019-07-30

**Authors:** Zhongyuan Wu, Jun Sang, Qian Zhang, Hong Xiang, Bin Cai, Xiaofeng Xia

**Affiliations:** 1Key Laboratory of Dependable Service Computing in Cyber Physical Society of Ministry of Education, Chongqing University, Chongqing 400044, China; 2School of Big Data & Software Engineering, Chongqing University, Chongqing 401331, China

**Keywords:** vehicle detection, YOLOv2, focal loss, anchor box, multi-scale

## Abstract

Vehicle detection is a challenging task in computer vision. In recent years, numerous vehicle detection methods have been proposed. Since the vehicles may have varying sizes in a scene, while the vehicles and the background in a scene may be with imbalanced sizes, the performance of vehicle detection is influenced. To obtain better performance on vehicle detection, a multi-scale vehicle detection method was proposed in this paper by improving YOLOv2. The main contributions of this paper include: (1) a new anchor box generation method Rk-means++ was proposed to enhance the adaptation of varying sizes of vehicles and achieve multi-scale detection; (2) Focal Loss was introduced into YOLOv2 for vehicle detection to reduce the negative influence on training resulting from imbalance between vehicles and background. The experimental results upon the Beijing Institute of Technology (BIT)-Vehicle public dataset demonstrated that the proposed method can obtain better performance on vehicle localization and recognition than that of other existing methods.

## 1. Introduction

Vehicle detection is one of the essential parts in computer vision, which aims to locate the vehicles and recognize the vehicle types. In recent years, vehicle detection has been applied in numerous fields, such as traffic surveillance, unmanned vehicle, gate monitoring and so on. However, due to the complicated background, the greatly varying illumination intensity, the occlusion problem and the small variations of each vehicle types, vehicle detection is still a challenging task and a hot research field of Artificial Intelligence (AI).

Numerous vehicle detection methods have been proposed, which can be divided into two categories: traditional machine learning methods and deep learning-based methods. For the traditional methods, Tsai et al. [1] used a new color transformation model to obtain the key vehicle colors and locate the candidate objects. Then, a multichannel classifier was adopted to recognize the candidate objects. For vehicle detection in video, Jazayeri et al. [2] thought that it should combine the temporal information of the features and the vehicle motion behaviors, which can compensate the complexity in recognizing vehicle colors and shapes. In Refs. [3,4,5], the Histogram of Oriented Gradient (HOG) method was applied to extract the vehicle features in the image, which have a lower false positive rate. In recent years, with the continuous improvement of computing power, deep learning [6] gradually becomes the main way for vehicle detection. The methods based on deep learning have surpassed the methods based on traditional methods [7,8,9,10] on detection and recognition performance. Different from designing feature by human beings, Ref. [11,12,13] proposed to learn features automatically with Convolutional Neural Network (CNN), which only needs large labeled vehicle images to train the network with supervision, while it does not need to design feature manually. The first systematic framework for object detection is R-CNN [14]. In R-CNN, the selective search algorithm [15] was used to generate the regions of interest, and then CNN was applied to recognize whether the region of interest is object or background. After R-CNN, numerous methods [16,17,18,19] adopted the same framework with slight changes for object detection. In Ref. [20] and Ref. [21], the Faster R-CNN was applied to detect the vehicles, which surpassed the previous methods. Ref. [22] developed an accurate vehicle proposal network to extract vehicle-like targets. Then, a coupled R-CNN method was proposed to extract the vehicle’s location and attributes simultaneously. To improve the detection speed, Redmon et al. [23] proposed YOLO, which was the first one to convert detecting object to regression and achieved end-to-end detection. Then, some other methods, such as SSD [24], YOLOv2 [25], YOLOv3 [26], etc., were proposed, which reduce the detection time greatly. Ref. [27] proposed an improved YOLOv2 for vehicle detection. k-means++ [28] was used to generate anchor boxes, instead of k-means [29], and the loss function was improved with normalization. In Ref. [30], by modifying the net resolution and depth of YOLOv2, the proposed model gave near Faster R-CNN performance at more than4 times speed. In object detection, usually the objects only occupy a small part of the image, while the majority in the image is background. The imbalance between objects and background can hinder object detection models from converging in a correct direction in the training stage. Therefore, to reduce the impact of the imbalance, Lin et al. [31] proposed Focal Loss to focus training on a sparse set of hard examples. Their experiments validated that the method can improve the detection result on end-to-end methods. 

To improve the accuracy of localization and recognition simultaneously, an improved vehicle detection method based on YOLOv2 was proposed in this paper. A new anchor box generation method was applied to enhance the network localization ability. In addition, Focal Loss was introduced in YOLOv2 to improve the network recognition ability.

## 2. Brief Introduction on YOLOv2

In YOLOv2 [25], the input image is divided into S × S grids. Each grid predicts the location information, the class probability and the object confidence for each anchor box. The location information includes x, y, w and h of the bounding box, where x, y represent the abscissa and ordinate of the center pixel of the bounding box, and w, h represent the length and height of the bounding box. The class probability indicates which class the object in the current grid is most likely to belong to. The object confidence represents the confidence that there exists objects in the current grid. Then, the Non-Maximum Suppression (NMS) is applied to remove the bounding boxes with low object confidence. Finally, the rest bounding boxes are decoded to obtain the detection boxes. In this section, YOLOv2 is introduced in brief, mainly including the generation of anchor boxes, the network structure and the loss function.

### 2.1. The Generation of Anchor Boxes

Anchor boxes were first proposed in Faster R-CNN [18], which aims to generate bounding boxes with a certain ratio instead of predicting the sizes of bounding boxes directly. The authors of YOLOv2 indicated that generating anchor boxes with manual design was absurd. Instead, they applied k-means cluster on training set to obtain better anchor boxes. 

When implementing k-means, instead of the traditional Euclidean distance, YOLOv2 adopted the Intersection over Union (IoU) distance to measure the closeness degree of two bounding boxes. The main reason is that, if the Euclidean distance is adopted, the anchor boxes with big size will produce more errors than those with small size. Consequently, by adopting the IoU distance, the errors will be irrelevant to the sizes of the anchor boxes. The distance of k-means in YOLOv2 can be expressed as Equation (1).
(1)d(box,centroid)=1−IOU(box,centroid)

### 2.2. The Network Structure

In recent years, most of the detection methods take VGG [7] or ResNet [8] as the base feature extractor. The authors of YOLOv2 argued that these networks were accurate and powerful, but they were needlessly complex. Therefore, Darknet19 was adopted as the backbone of YOLOv2, which has less parameters and may obtain better performance than VGG and ResNet. The network of YOLOv2 is shown in Figure 1.

As shown in Figure 1, the network includes 32 layers, which have 19 convolutional layers and five maxpooling layers. Similar to VGG, 3 × 3 convolutional layer is used to double the channel of feature maps after each pooling layers, and 1 × 1 convolutional layer is used to halve the channels and fuse the features. Feature fusion is applied on the feature maps from the direct path and path (a) in Figure 1, which can retain the features from the shallower layer to improve the ability of detecting the small objects.

### 2.3. The Loss Function

In YOLOv2, multi-part loss function is adopted, which includes the bounding box loss, the confidence loss and the class loss. The bounding box loss includes the coordinate loss and the size loss of the bounding box. The confidence loss includes the confidence loss of bounding box with objects and without objects. The class loss is calculated by softmax to obtain the class probability. The loss function in YOLOv2 can be expressed as Equation (2).
(2)$$\begin{array}{c} \lambda_{coord}\sum_{i=0}^{S^{2}}\sum_{j=0}^{B}I_{ij}^{obj}[{\left(x_{i}-{\hat{x}}_{i}\right)}^{2}+{\left(y_{i}-{\hat{y}}_{i}\right)}^{2}]\\ +\lambda_{coord}\sum_{i=0}^{S^{2}}\sum_{j=0}^{B}I_{ij}^{obj}[{\left(\sqrt{w_{i}}-\sqrt{{\hat{w}}_{i}}\right)}^{2}+{\left(\sqrt{h_{i}}-\sqrt{{\hat{h}}_{i}}\right)}^{2}]\\ +\sum_{i=0}^{S^{2}}\sum_{j=0}^{B}I_{ij}^{obj}{\left(C_{i}-{\hat{C}}_{i}\right)}^{2}\\ +\lambda_{noobj}\sum_{i=0}^{S^{2}}\sum_{j=0}^{B}I_{ij}^{noobj}{\left(C_{i}-{\hat{C}}_{i}\right)}^{2}\\ +\sum_{i=0}^{S^{2}}I_{i}^{obj}\sum_{c\in classes}\left(p_{i}(c)-{\hat{p}}_{i}(c\right){)}^{2} \end{array}$$

As shown in Equation (2), $x_{i}$ and $y_{i}$ denote the center coordinates of the box relative to the current grid bounds in the *i*-th grid. $w_{i}$ and $h_{i}$ denote width and height of the bounding box relative to the whole image in the *i*-th grid. $C_{i}$ denotes the confidence of the bounding box in the *i*-th grid. $p_{i}(c)$ denotes the class probability of the bounding box in the *i*-th grid. ${\hat{x}}_{i}$, ${\hat{y}}_{i}$, ${\hat{w}}_{i}$, ${\hat{h}}_{i}$, ${\hat{C}}_{i}$, ${\hat{p}}_{i}(c)$ denote the corresponding predicted values of $x_{i}$, $y_{i}$, $w_{i}$, $h_{i}$, $C_{i}$, $p_{i}(c)$. $S^{2}$ denotes the S × S grids. B denotes the bounding boxes. $\lambda_{coord}$ denotes the weight of the coordinate loss and $\lambda_{noobj}$ denotes the weight of the loss of bounding boxes without objects. $I_{i}^{obj}$ denotes whether the object is on the *i*-th grid or not and $I_{ij}^{obj}$ denotes whether the *j*-th box predictor in the *i*-th grid is “responsible” for that prediction or not.

In the loss function, the first line is to compute the coordinate loss, the second line is to compute the bounding box size loss, the third line is to compute the bounding box confidence loss containing objects and the last line is to compute the bounding box confidence loss not containing objects. To prevent the sizes of the bounding boxes from making a significant impact on the loss, the square roots of width and length of bounding boxes are applied to decrease their magnitudes. Since usually only a few bounding boxes with object exist in the real pictures, the confidence loss of bounding box with object is much smaller than the other losses. Consequently, the weighted method is applied to balance the different kinds of losses. Usually, $\lambda_{coord}$ is set as 5 and $\lambda_{noobj}$ is set as 0.5 to balance each loss. Otherwise, each loss may result in different contributions to the total loss, which can cause some losses ineffective for network training.

## 3. The Proposed Method

The proposed vehicle detection method was based on YOLOv2, in which a new anchor box generation method was proposed and Focal Loss was introduced in YOLOv2. In this section, these two improved points will be introduced in detail.

### 3.1. The Generation of Anchor Boxes

The quality of anchor boxes is important for the end-to-end detection methods. It is efficient by using k-means or k-means++ to generate anchor boxes. However, the anchor boxes generated by such methods are usually suitable to the common sizes of objects, while the sizes of the generated anchor boxes may be far away from those of the objects with usual sizes. By analyzing the detection boxes of YOLOv2, we found that the common sizes of the vehicles can be predicted well, while some unusual sizes of vehicles, such as the size of a truck which is usually much bigger than that of the common vehicles, were predicted terribly. As shown in Figure 2, the black boxes denote the ground truths and the other color boxes denote the detection boxes predicted by YOLOv2. It is obvious that the sedan was detected well and the minivan was detected terribly, since the size of minivan is unusual and the size of sedan is common.

To improve the localization accuracy, it is better to generate the anchor boxes to match most sizes of the ground truths, instead of only matching the common sizes of the ground truths. Therefore, a clustering method called Rk-means++ was proposed in this paper. As shown in Figure 3, in Rk-means++, 2 ratios regarding the width and the length of the ground truths were obtained by applying k-means++. Then, we applied a certain proportion to two ratios and generate anchor boxes with different scale. Compared with the methods of clustering anchor boxes directly, such as k-means and k-means++, Rk-means++ generates anchor boxes on different hierarchies, which may match most sizes of the ground truths much better. In our experiments, the proportion was set as 1:2:4:6:8:10 and six anchor boxes were obtained.

### 3.2. Focal Loss

For vehicle detection, usually the vehicles are only a small part of the whole image, while the majority in the image is background. Consequently, thousands of candidate bounding boxes will be generated in one image with YOLOv2, while only a few of them include vehicles. Obviously, the problem of imbalance between the positives (i.e., vehicles) and the negatives (i.e., backgrounds) is serious. In the training stage of YOLOv2, because of the large quantity of candidate bounding boxes not containing object, the loss will be overwhelmed by them and a few quantity of candidate bounding boxes containing object cannot influence the loss effectively. The imbalance of quantity will make training inefficient since most candidate bounding boxes tend to be easy negative, which is useless for CNN learning. The numerous easy negatives will overwhelm some important training examples, which leads the model to converge in a wrong direction. Therefore, in order to decrease the imbalance between the positives and negatives, Focal Loss [31] was introduced in YOLOv2 to detect the vehicles.

Before Focal Loss was introduced, the cross entropy loss is commonly employed to calculate the classification loss, which is shown in Equation (3) for binary classification.
(3)$$L_{ce}=-y\log y^{\prime }-(1-y)\log(1-y^{\prime })=\{\begin{array}{cc} -\log y^{\prime }, & y=1\\ -\log(1-y^{\prime }), & y=0 \end{array}$$

In Equation (3), y denotes the ground truth, which is 1 for positive and 0 for negative. $y^{\prime }$ denotes the predicted value ranging from 0 to 1. For the positives, the higher the predicted probability is, the smaller the cross entropy loss is. For the negatives, the lower the predicted probability is, the smaller the cross entropy loss is. Consequently, it is inefficient to train with the iteration of numerous easy examples. More seriously, the model may not be optimized to a good state.

Focal Loss is based on cross entropy loss, which aims to reduce the weights of easy examples in loss and make the model focus on distinguishing the hard examples. Focal Loss can be expressed as Equation (4).
(4)$$L_{fl}=\{\begin{array}{cc} -\alpha {\left(1-y^{\prime }\right)}^{\gamma }\log y^{\prime }, & y=1\\ -(1-\alpha ){y^{\prime }}^{\gamma }\log(1-y^{\prime }), & y=0 \end{array}$$

As shown in Equation (4), corresponding to cross entropy loss, 2 factors, i.e., γ and α, are added. γ is used to reduce the loss of easy examples. For instance, by setting γ as 2, the loss will be smaller than the cross entropy loss if an easy positive with predicted value 0.95 or an easy negative with predicted value 0.05. The factor γ makes the training more focus on the hard examples and reduce the impact of easy examples. α is used to balance the weights of positives and negatives, which can control the decline rate for the weights of examples. Within the limited numbers of experiments, we found that setting γ as 2 and setting α as 0.25 can obtain the best accuracy.

## 4. Experiments

The implementation of our experiments was based on Darknet, which is a light deep learning framework and provided by the author of YOLOv2. The experiments were conducted on the GPU server, which includes 8 pieces of GPUs, 24 G video memory and 64 G memory. The experimental platform was equipped with 64-bit ubuntu14.04, Opencv2.7.0 and CUDA Toolkit8.0. 

### 4.1. Dataset

The experiments were conducted on BIT-Vehicle [32] public dataset provided by Beijing Institute of Technology. The BIT-Vehicle dataset contains 9580 images, including 6 vehicle types, i.e., bus, microbus, minivan, sedan, truck and suv. The numbers of each vehicle type are 558, 883, 476, 5922, 822 and 1392. As shown in Figure 4, the BIT-Vehicle dataset includes day scene and night scene, and the images are seriously influenced by the variation of illumination intensity.

In our experiments, the BIT-Vehicle dataset was divided into training set and test set with the ration of 8:2, namely, 7880 images were used for training and 1970 images were used for testing.

### 4.2. Implementation

In our experiments, the size of input image was 416 × 416. The initial learning rate was set to be 0.001. The model was trained by 120 epochs and the learning rate was decreased 10 times when the iteration reached 60 and 90 epochs, respectively. The size of batch was set to be 8 and the momentum was set as 0.9. In order to make the model detect well for the images with different sizes, the multi-scale training trick was applied in our experiments. A new size of input image was selected randomly for training every 10 epochs. Since the factor of down sampling is 32 in YOLOv2, the sizes of the randomly selected input images all were multiple of 32, i.e., the maximum size was 608 × 608 and the minimum size was 352 × 352. Other hyperparameters followed as those in YOLOv2.

### 4.3. Experimental Results and Analysis

The proposed method was compared with YOLOv2_Vehicle [27] and the other methods for vehicle detection. In addition, all of these methods were implemented on our platform, so the results of these methods were obtained under the same experimental environment. The experimental results are shown in Table 1. 

As shown in Table 1, Class Average Precision (AP) denotes the recognition performance for each vehicle class, mean Average Precision (mAP) denotes the average recognition performance and IoU denotes the localization performance. It can be seen that mAP and IoU obtained with our proposed method are 97.30% and 92.97%, respectively. Specifically, the mAP of our method is higher than those of other methods by nearly 1 percentage point and the IoU is higher than those of other methods by nearly 1 percentage points. Significantly, except the speed and AP of bus, microbus, all the metrics obtained by our proposed method are better than those of the others. For the AP of bus and microbus, our method is not the best one. The reason may be that the different models will focus on different classes due to the random initial parameters, the network architecture or the anchor boxes being different. Consequently, it is acceptable that there are two class APs slightly smaller than others. For the speed, our method is not the best one, which may result from the fact that the more candidate bounding boxes were proposed before the NMS, which slow down the speed lightly. In general, our proposed method showed better performance than others on the accuracy of localization and recognition, while maintaining the speed.

Figure 5 showed the detection results of our proposed method. Whether it was daytime or night, the abilities of localization and recognition were not affected. Whether the vehicle is with small size or with large size, our method can locate and recognize them as well. In addition, it can be seen from the second picture and last picture of the second column in Figure 5, though the vehicle was incomplete, they can be detected accurately. Consequently, regardless of the variation of illumination intensity, the variation of vehicle sizes or the vehicle completeness, our method can recognize and locate the vehicles accurately. It demonstrates the strong robustness of our proposed method.

### 4.4. The Validation of Rk-means++

In order to validate the effect of Rk-means++ for YOLOv2 in vehicle detection, numerous experiments were conducted to compare the performance of different methods with different anchor box generation methods. In the experiments, YOLOv2 and YOLOv3 were adopted as the basic model. The experimental results are shown in Table 2.

As shown in Table 2, it is obvious that the IoU with Rk-means++ are the best with YOLOv2 and YOLOv3.The reason could be that each size of vehicle with different scales can be matched with one of the anchor boxes, since the anchor boxes generated by Rk-means++ are on different hierarchies. However, the mAP with Rk-means++ is slightly lower than that of others. This may result from the fact that more candidate bounding boxes were proposed with Rk-means++, which makes the detection precision drop slightly. In general, with Rk-means++, the model can obtain better localization performance while the precision is slightly lower.

### 4.5. The Validation of Focal Loss

In order to validate the effect of Focal loss for YOLOv2 in vehicle detection, YOLOv2 was adopted as the basic model with different anchor box generation methods. The experimental results are shown in Table 3, where ‘Wo’ denotes ‘Without’ and ‘FL’ denotes ‘Focal Loss’.

As shown in Table 3, k-means, k-means++ and Rk-means++ were applied to compare the performance without or with Focal Loss. It is obvious that all the YOLOv2 models with Focal Loss will surpass those without Focal Loss regardless of the different anchor box generation methods, which indicates that Focal Loss can reduce the influence on imbalance of positives and negatives. It also validates that Focal Loss is useful for improving both mAP and IoU.

## 5. Conclusions

In this paper, a multi-scale vehicle detection method based on YOLOv2 was proposed to improve the accuracy of localization and recognition simultaneously. The main contributions of this paper include:(1)By analyzing the detection boxes of YOLOv2, it was found that the common sizes of the vehicles can be predicted well, but some unusual sizes of vehicles were predicted terribly. Therefore, to adapt most sizes of vehicles with different scales, instead of clustering the anchor boxes directly, a new anchor box generation method, Rk-means++, was proposed.(2)By analyzing the loss of the training stage, it was found that most of the loss was from the irrelevant background, which is due to the imbalance between the vehicles and background in each image. Therefore, the Focal Loss was introduced in YOLOv2 for vehicle detection to reduce the impact of easy negatives (background) on loss and make the loss focus on the hard examples.(3)The experiments conducted on the BIT-Vehicle public dataset demonstrated that our improved YOLOv2 obtained superior performance than others on vehicle detection.(4)By conducting numerous comparison experiments, the effectiveness of Rk-means++ and Focal Loss was validated.

Although the method proposed in this paper shows good performance on vehicle detection, the scales of data and the types of vehicle are relatively few, which cannot show the distinctions of different methods well. In the future work, we will collect more vehicle data and try to improve the accuracy on the larger vehicle dataset.

## Figures and Tables

**Figure 1 sensors-19-03336-f001:**
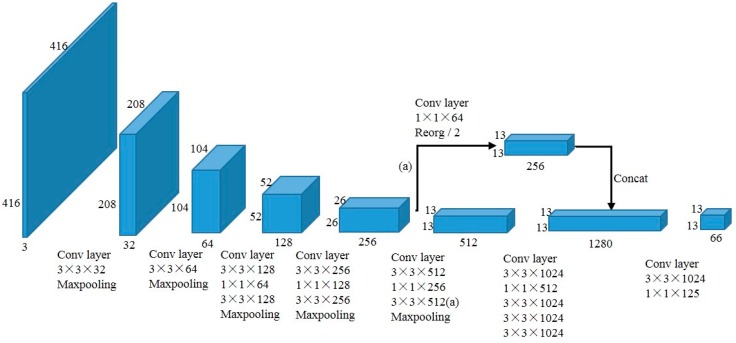
The network of YOLOv2.

**Figure 2 sensors-19-03336-f002:**
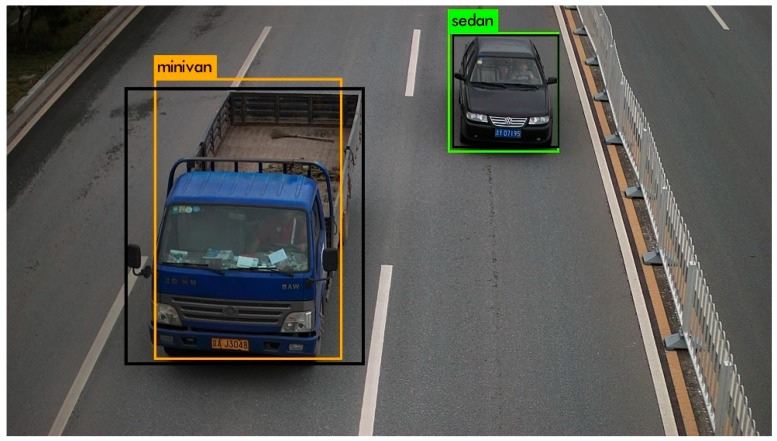
The detection boxes generated by YOLOv2.

**Figure 3 sensors-19-03336-f003:**
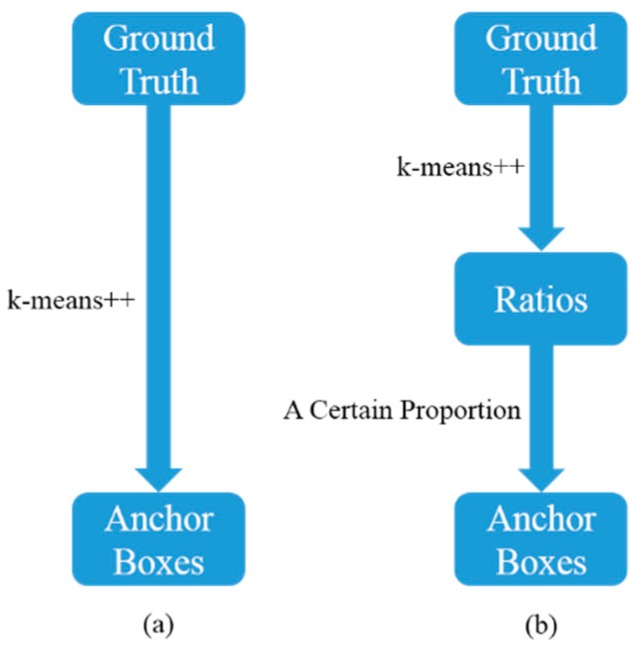
The computing procedures of k-means++ and Rk-means++. (**a**) k-means++; (**b**) Rk-means++. The anchor boxes obtained by Rk-means++ ensures that each size of the vehicle with different scale can be matched with one of the anchor boxes. In other words, the proposed anchor box generation method can enhance the robustness of YOLOv2 for different scales of vehicles and improve the localization accuracy.

**Figure 4 sensors-19-03336-f004:**
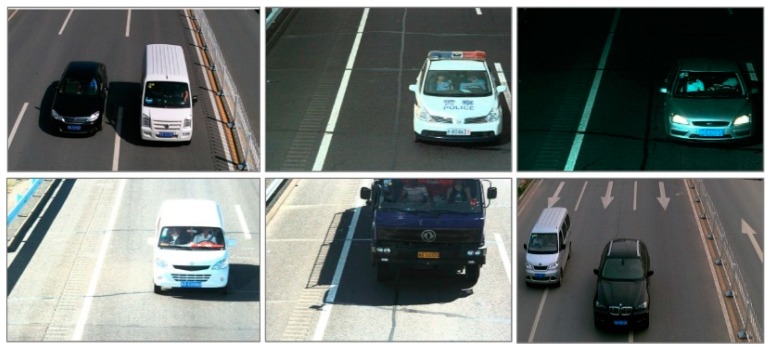
Some images in BIT-Vehicle dataset.

**Figure 5 sensors-19-03336-f005:**
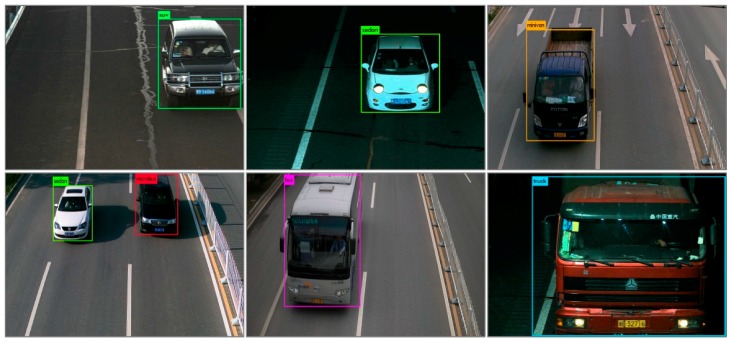
Examples of detection results.

**Table 1 sensors-19-03336-t001:** Experimental results for different methods.

Method	The Class AP (%)						mAP (%)	IoU (%)	Speed (s)
	Bus	Microbus	Minivan	Sedan	SUV	Truck			
YOLOv2 [25]	98.34	95.03	91.11	97.42	93.62	98.41	95.65	90.44	0.0496
YOLOv2_Vehicle [27]	98.42	97.04	95.02	97.37	93.73	97.80	96.56	91.06	0.0486
YOLOv3 [26]	98.65	96.98	94.04	97.65	94.36	98.17	96.64	88.50	0.1100
SSD300 VGG16 [24]	97.97	**97.98**	90.28	97.15	91.25	97.75	93.75	91.60	**0.0440**
Faster R-CNN VGG16 [18]	**99.05**	93.75	91.38	98.14	94.75	98.17	95.87	92.19	0.4257
Our Method	98.86	96.63	**95.90**	**98.23**	**94.86**	**99.30**	**97.30**	**92.97**	0.0522

**Table 2 sensors-19-03336-t002:** Experimental results on comparing different anchor box generation methods.

Method		The Class AP (%)						mAP (%)	IoU (%)
		Bus	Microbus	Minivan	Sedan	SUV	Truck		
YOLOv2	k-means	98.34	95.03	91.11	97.42	93.62	98.41	95.65	90.44
	k-means++	98.60	96.29	93.16	97.47	93.72	98.15	**96.23**	91.05
	Rk-means++	98.68	96.66	91.50	97.48	93.59	97.33	95.88	**92.18**
YOLOv3	k-means	98.65	96.98	94.04	97.65	94.36	98.17	**96.64**	88.50
	k-means++	98.70	96.44	93.80	97.66	94.76	97.78	96.52	88.86
	Rk-means++	98.32	97.08	92.65	97.70	94.27	97.96	96.33	**90.49**

**Table 3 sensors-19-03336-t003:** Experimental result on comparing YOLOv2 with Focal Loss (FL).

YOLOv2		The Class AP (%)						mAP (%)	IoU (%)
		Bus	Microbus	Minivan	Sedan	SUV	Truck		
k-means	Wo FL	98.34	95.03	91.11	97.42	93.62	98.41	95.65	90.44
	With FL	98.34	96.80	94.81	98.20	95.34	99.32	**97.13**	**92.06**
k-means++	Wo FL	98.60	96.29	93.16	97.47	93.72	98.15	96.23	91.05
	With FL	98.48	97.44	96.03	98.24	95.68	99.17	**97.51**	**92.30**
Rk-means++	Wo FL	98.68	96.66	91.50	97.48	93.59	97.33	95.88	92.18
	With FL	98.86	96.63	95.90	98.23	94.86	99.30	**97.30**	**92.97**

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
