# Peer review of "Multi-Scale Vehicle Detection for Foreground-Background Class Imbalance with Improved YOLOv2"

_sensors, 2019, doi:10.3390/s19153336_

Round 1

Reviewer 1 Report

In this paper, a multi-scale vehicle detection method based on YOLOv2 was proposed to improve the accuracy of localization and recognition simultaneously. The authors propose Rk-means method for generating anchor shapes and adopt the Focal Loss to deal with the class-imbalance problem better. The structure of the paper is good. But I have some concerns:

1. The process of Rk-means is not well presented. I suggest the authors use equations, tables, or figures to show its computing procedure and the difference between it and the methods like K-means.

2. The necessary of using Focal Loss is not well discussed. The anthors think the loss value of the training process mainly comes from the easy nagetives, but they have not shown that in the paper.

3. The related works are too old. Many recent works on object detection (like SSD, FPN, Fast RCNN, PleeNet, etc. )  are not compared in the experimental section.

4. The paper is about vehical detection. However, the paper does not mention the other recent vehical detection methods in the introduction and experimental section.

5. There are some grammar errors. Like "maybe with" -> "may be with" in the abstract;

"proposed to improving"->"proposed to improve" in the conclusion. The authors should use the grammar checking tools to check their presentations.

Reviewer 2 Report

In Line 106, the definition of the loss function should be explained detaily, why in this form and how to determine the paramters and also different effect uder different chosen parameters. 

 The process of the proposed method in PART 3 should be described in a flow chart to make it clear.

Experimental results are inadequte, the light illumination, the object obstacle, the vehicle speed and some other factors should be considered when assess the performance.

Reviewer 3 Report

The topic of the paper ‘Multi-scale Vehicle Detection for Foreground background Class Imbalance with Improved YOLOv2’ is very interesting. To obtain better performance on vehicle detection, a multi-scale vehicle detection method was proposed in this paper by improving YOLOv2. A new anchor box generation method was applied to enhance the network localization ability. In addition, Focal Loss was introduced in YOLOv2 to improve the network recognition ability.

The manuscript is well written, and in my opinion meet all the standards for publication in this journal.

Round 2

Reviewer 1 Report

I have no further comments to this paper and recommend accepting it after the authors correct the format of their paper. 

Author Response

N